# Nickel-catalyzed $\beta$-arylation and benzylation of 2′-hydroxychalcones to access warfarin analogues
Bo-Cheng Tang[1], Xiaochun Su[1,2], Jiang Liu[1,2], Xiao Yang[2,3] ✉ & Cong Ma [1,2] ✉

The development of new methods for forming carbon–carbon bonds is essential for advancing the synthesis of biologically active molecules. Achieving high selectivity in these reactions remains a significant challenge in organic chemistry. Here we show that a nickel-catalyzed $\beta$-arylation and benzylation of 2′-hydroxychalcones enables the efficient synthesis of chalcone derivatives. This transformation is directed by the substrate's intrinsic hydroxy group, resulting in high chemoselectivity and avoiding unwanted byproducts. The resulting chalcone derivatives can be converted in one step into a series of 3-functionalized 4-hydroxycoumarin compounds. These compounds demonstrate excellent anticoagulant effects in animal studies, with some showing greater activity than the widely used drug warfarin. This approach offers a promising strategy for developing therapeutic agents and functional materials based on the chalcone structure.

Transition-metal-catalyzed hydroarylation is a crucial technique for alkene functionalization, facilitating efficient and selective C-C bond formation—a fundamental step in constructing complex organic molecules. This process has been under investigation since the 1980s, with early efforts focusing on capturing the Pd intermediate formed after migratory insertion in the Heck reaction by introducing an exogenous hydrogen source to prevent alkene product formation[1–4]. However, controlling the selectivity of these reactions has been challenging, often leading to mixtures of various products. Currently, diverse strategies employing different transition metals enable hydroarylation of alkenes. For instance, Pd-catalyzed reactions can yield hydroarylation products by incorporating effective hydride sources or blocking groups to inhibit active $\beta$-H elimination (Scheme 1a)[5–11]. Additionally, specific alkene structures, such as norbornene (rigid structure)[12,13], styrene[14–16], and 1,3-dienes ($\pi$-allyl metal complex)[17,18], can undergo further functionalization. Due to inherent differences between Ni and Pd, Ni is less susceptible to $\beta$-H elimination, offering unique opportunities in this field[19]. Researchers have developed numerous important methods for alkene hydroarylation[20–28], with the introduction of aminoquinoline structures as directing groups proving highly effective[29–33]. Given the significant application of alkene hydroarylation in pharmaceuticals and natural product synthesis[34–38], developing efficient and broad-spectrum methods for these reactions remains highly desirable.

Substrate-directable reactions employ internal functional groups to enable, accelerate, or control the stereo- or regiochemical outcomes of reactions[39]. These reactions often utilize inherent, simple functional groups, such as hydroxy or amino groups, eliminating the need for the pre-introduction and subsequent removal of additional directing groups[40,41]. This makes substrate-directable reactions a valuable strategy in organic reaction design. Numerous substrate-directable reactions have been reported, including cyclopropanation[42,43], epoxidation[44], nucleophilic addition[45–47], hydrogenation[48], C-H activation[49–54], di-functionalization[55,56], and hydroarylation of alkenes[57–59]. For instance, Wang et al. demonstrated a Mn-catalyzed hydroarylation of alkenes using an internal amide group in the substrate as a directing strategy (Scheme 1b)[60]. Inspired by these studies, we hypothesized that a functional group within the substrate that could direct the reaction and serve a valuable role in subsequent synthetic steps would be highly advantageous. Herein, we present a Ni-catalyzed $\beta$-arylation/benzylation of chalcones, enabled by the pivotal hydroxy group (Scheme 1c).

## Results

### Optimization of reaction conditions

We selected 2'-hydroxychalcone and iodobenzene as model substrates for our study and conducted a comprehensive investigation of various parameters affecting the reaction (Table 1). The type of Ni catalyst was found to significantly influence the reaction yield. For example, neither Ni(cod)$_2$ nor Ni(acac)$_2$ produced any detectable product, whereas NiI$_2$ increased the yield to 56%, outperforming Ni(DME)Br$_2$ (Entries 1–5). DMF emerged as the

[1]State Key Laboratory of Chemical Biology and Drug Discovery, and Department of Applied Biology and Chemical Technology, The Hong Kong Polytechnic University, Kowloon, Hong Kong SAR, China. [2]Marshall Research Centre for Medical Microbial Biotechnology, The Hong Kong Polytechnic University, Kowloon, Hong Kong SAR, China. [3]Department of Microbiology, The Chinese University of Hong Kong, Prince of Wales Hospital, Shatin, Hong Kong SAR, China. ✉e-mail: xiaoyang@cuhk.edu.hk; cong.ma@polyu.edu.hk

## Table 1 | Optimization of Reaction Conditions[a,b]

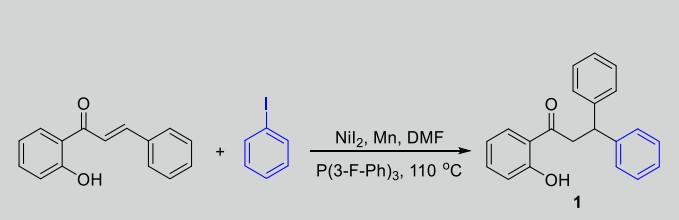

| Entry | Deviation of standard condition | Yield[b] (%) |
|---|---|---|
| **1** | **none** | **78** |
| 2 | No ligand | 56 |
| 3 | Ni(cod)₂ instead of NiI₂ | n.d. |
| 4 | Ni(acac)₂ instead of NiI₂ | n.d. |
| 5 | Ni(DME)Br₂ instead of NiI₂ | 48 |
| 6 | NMP instead of DMF | 42 |
| 7 | THF at 80 °C instead of DMF | 46 |
| 8 | n-BuOH instead of DMF | n.d. |
| 9 | toluene instead of DMF | n.d. |
| 10 | dioxane instead of DMF | n.d. |
| 11 | DME instead of DMF | n.d. |
| 12 | 1,10-Phen instead of P(3-F-Ph)₃ | trace |
| 13 | PPh₃ instead of P(3-F-Ph)₃ | 65 |
| 14 | P(O)Ph₃ instead of P(3-F-Ph)₃ | 58 |
| 15 | P(3-Cl-Ph)₃ instead of P(3-F-Ph)₃ | 76 |
| 16 | PPh₂Bn instead of P(3-F-Ph)₃ | 75 |
| 17 | PPh₂Cy instead of P(3-F-Ph)₃ | 72 |
| 18 | No Mn | n.d. |
| 19 | 1 eq of Mn | 33 |

The bold values demonstrate the optimal yield obtained under the reaction conditions demonstrated in the table scheme.

[a]Reaction conditions: 2'-hydroxychalcone (0.5 mmol), Iodobenzene (1.5 mmol), NiI₂ (10 mol%), P(3-F-Ph)₃ (22 mol%), Mn (3.0 eq), DMF (4.0 ml), 110 °C, 16 h. [b]Isolated yield.

optimal solvent, as substituting it with toluene, dioxane, n-BuOH, or DME inhibited the reaction, and both NMP and THF resulted in a notable decrease in yield (Entries 6–11, see Table S1 for more details). Although a ligand was not strictly necessary for this model reaction, the addition of an appropriate ligand further enhanced the yield and minimized byproducts from direct alkene reduction. Optimization revealed that nitrogen-containing ligands inhibited the reaction, while *meta*-substituted triphenylphosphine derivatives performed favorably (Entries 12–17). Additionally, both Mn and the Ni catalyst were essential, and reducing the equivalents of iodobenzene, or Mn below 2.5 eq. significantly decreased the yield. Other types of reducing reagents were also explored, including Zn, HSiEt₃, PhSiH₃, and HCOOK. However, none of these alternatives provided a higher yield than Mn (see Table S2 for more details). Under the optimized conditions (Entry 1), product 1 was obtained with excellent selectivity and an isolated yield of 78%.

### Scope of the reaction

We then conducted a systematic study on the substrate scope of this reaction. Initially, various substituents at the *para*-position of the iodobenzene ring were evaluated (Scheme 2). The reaction demonstrated good tolerance for common electron-donating, electron-withdrawing, and halogen groups, yielding the corresponding products (1–8) with moderate to good yields (54–78%). Additionally, *meta*- and *ortho*-substituted, as well as poly-substituted groups, were investigated, and the reaction proceeded smoothly to afford the desired products (9–12). Notably, functional groups, such as alcohol, aldehyde, cyano, and amide were compatible with the reaction,

offering opportunities for potential subsequent synthetic transformations (13–17). Furthermore, a variety of fused heterocycles, including benzo-dioxane, benzodioxole, naphthalene, and phenanthrene, were examined and yielded the corresponding products in good yields (18–19, 21–24). We also assessed the compatibility of several alkenyl halides, such as (2-bromovinyl)benzene, although these substrates generally resulted in significant Z/E mixtures, indene was compatible with the reaction, providing the corresponding product in a moderate yield of 53% (25). Additionally, aliphatic substituents and deuterated iodobenzene were well-tolerated under these reaction conditions (20, 26). The structure of compound 1 was confirmed by single-crystal X-ray diffraction analysis.

[a]Reaction conditions: alkene (0.5 mmol), aryl iodide (3.0 equiv), Mn (3.0 equiv), NiI₂ (10 mol%), P(3-F-Ph)₃ (22 mol%), DMF (4.0 mL), 110 °C, 16 h. [b]Isolated yield.

To further expand the scope of this synthetic method, we explored the construction of a C(sp³)-C(sp²) bond under the same optimized reaction conditions. Encouragingly, we achieved efficient and highly selective benzylation using benzyl bromide as the benzylating reagent. We then investigated the substrate scope of this hydro-benzylation, focusing on functional groups not present in the aryl iodide derivatives to maximize the demonstration of functional group tolerance (Scheme 3). Initially, *tert*-butyl, phenyl, and fluoro-substituted groups were successfully converted into the corresponding products with moderate to good yields (27–33). Hydroxy, sulfone, sulfonamide, and 3,4-disubstituted groups also showed good tolerance in this process (34–37). Although steric hindrance slightly reduced the yield, compound 38 was successfully obtained. Functional groups, such as acetyl and protected amines, which are useful for further transformations, were achieved with acceptable yields (39–40). We then assessed the compatibility of bromomethyl compounds containing fused rings and heterocycles, finding that naphthyl, thiophene, and furan exhibited good reactivity, producing the desired products (41–43). Notably, substituents, such as methyl, cyclobutyl, and lactone at the benzylic position were also evaluated, and the target products were successfully obtained (44–46). The structure of compound 29 was confirmed by single-crystal X-ray diffraction analysis. We have also tested a series of alkyl bromides. Unfortunately, these experiments resulted in the formation of highly complex mixtures, rather than the desired products. This outcome suggests that the current protocol may not be suitable for alkyl bromides, possibly due to competing side reactions or lower selectivity.

[a]Reaction conditions: alkene (0.5 mmol), bromomethyl derivative (3.0 equiv), Mn (3.0 equiv), NiI₂ (10 mol%), P(3-F-Ph)₃ (22 mol%), DMF (4.0 mL), 110 °C, 16 h. [b]Isolated yields based on alkene.

Finally, we examined the diversity of aromatic rings on both sides of the olefin (Scheme 4). The reaction demonstrated good efficiency and tolerance for various substituents, including alkyl and alkoxy groups (47–48), halogen-containing groups (49–52), nitrogen-containing groups (53–54), and heterocycles (55–57), yielding the corresponding products efficiently. The structure of compound 56 was confirmed by single-crystal X-ray diffraction analysis.

[a]Reaction conditions: alkene (0.5 mmol), bromomethyl derivative (3.0 equiv), Mn (3.0 equiv), NiI₂ (10 mol%), P(3-F-Ph)₃ (22 mol%), DMF (4.0 mL), 110 °C, 16 h. [b]Isolated yields based on alkene.

Overall, this synthetic method efficiently synthesized chalcone derivatives and demonstrated excellent functional group compatibility. Chalcone is a highly regarded scaffold known for its diverse bioactivities[61] and its use in the synthesis of functional materials[62]. The ability to access chalcone derivatives through this method may offer significant potential for a wide range of applications.

### Synthetic applications

We further investigated the practicality of this synthetic method. Initially, we conducted a standard reaction on a 4.5 gram scale, achieving an isolated yield of 70% for the target compound after column chromatography (Scheme 5a). We then employed diethyl carbonate as the carbonyl source

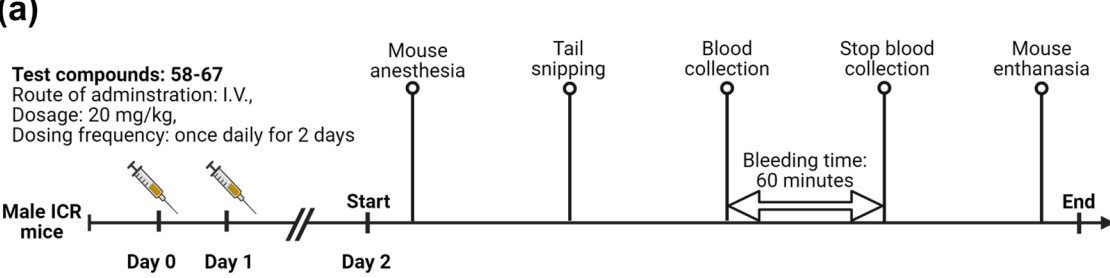

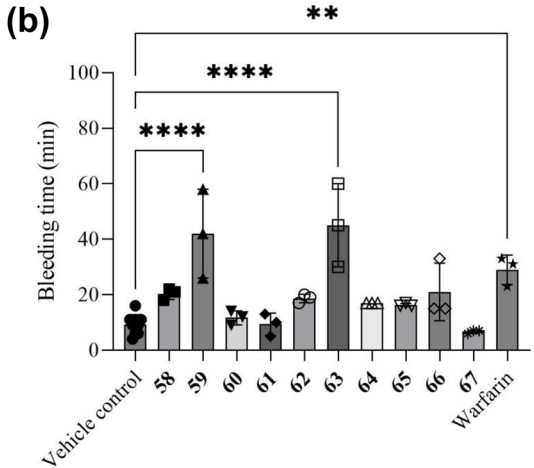

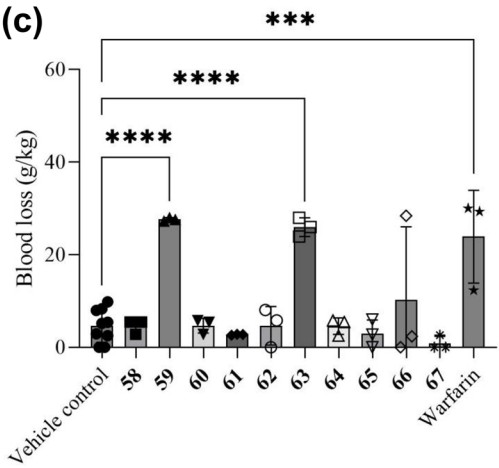

**Fig. 1 | Evaluation of anticoagulant efficacy of compounds 58–67 in ICR mice.** **a** Schematic diagram of dosing and blood sampling in the in vivo bleeding model. Male ICR mice (n = 3 per group) were administered with compounds 58–67 or warfarin via I.V. injection once daily for two consecutive days, respectively. Bleeding time and blood loss were assessed on Day 2, 24 h after the final dose. **b** Bleeding time in ICR mice following treatment with compounds 58–67 or warfarin, compared to the vehicle control group. Bleeding time was defined as the time elapsed until bleeding ceased. **c** Blood loss in ICR mice following treatment with compounds 58–67 or warfarin, expressed as the ratio of blood weight collected to the mouse body weight. Data is presented as mean ± SD (n = 3 biologically independent mice per group). Data was analyzed using one-way ANOVA (two-sided) to determine the significant differences among groups. $**p < 0.01$, $***p < 0.001$, $****p < 0.0001$.

**Scheme 1 | Selected examples of transition-metal-catalyzed hydroarylation of alkenes. a** Reductive Mizoroki-Heck reaction. **b** Internal amide group-assisted hydroarylation. **c** This work: Hydroarylation/benzylation of chalcones enabled by the hydroxy group.

(a) Reductive Mizoroki-Heck reaction

(b) Internal amide group assisted hydroarylation

(c) This work: Hydroarylation/benzylation of chalcones enabled by the hydroxy group

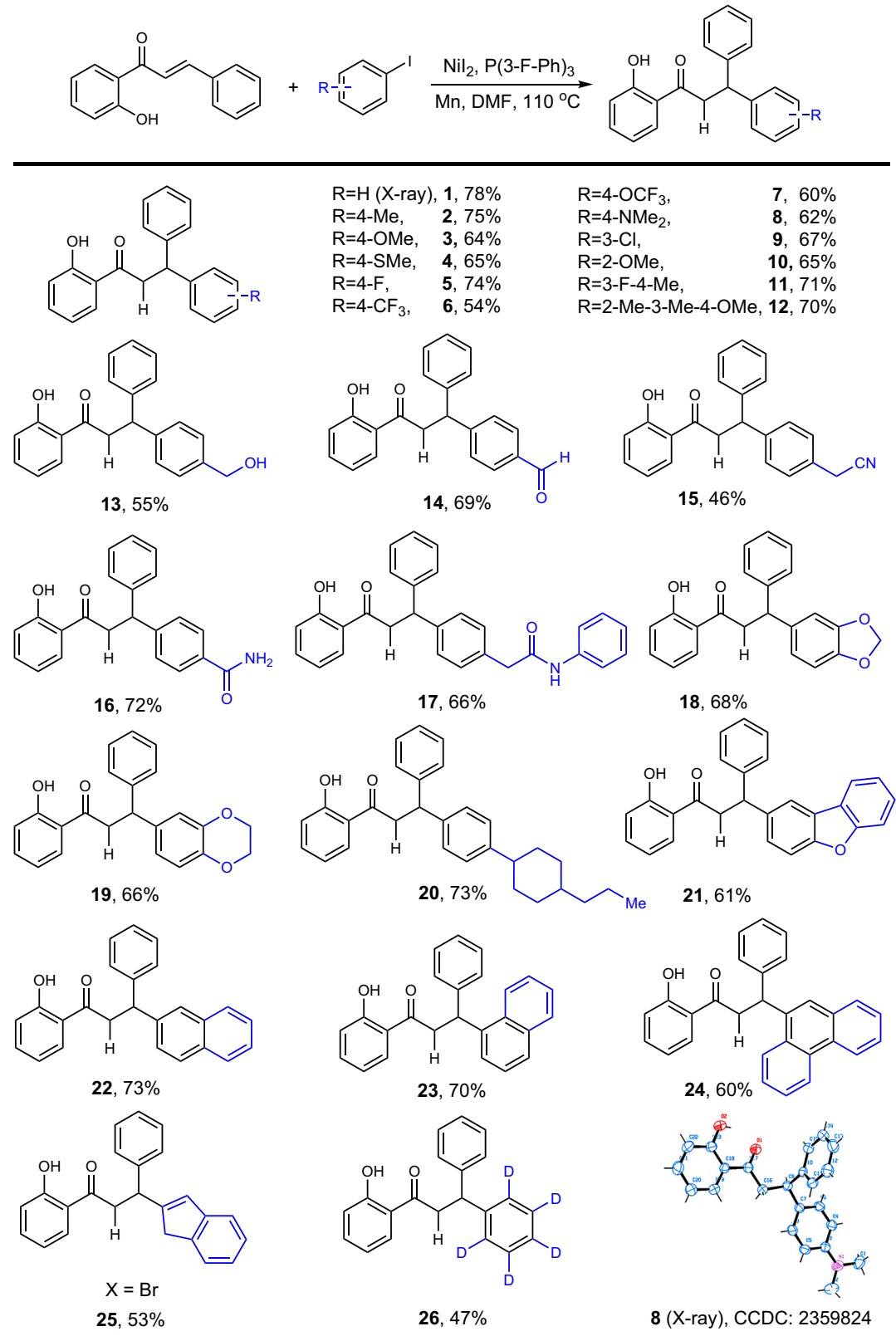

**Scheme 2 | Scope of aryl iodides**[a,b].

(Scheme 5b) and dibromomethane as the C1 source (Scheme 5c) to facilitate further cyclization reactions. These reactions led to the synthesis of a series of 3-functionalized 4-hydroxycoumarin and chroman derivatives (58–68). Notably, compounds 58–67 possess the same scaffold as warfarin, a well-known 3-functionalized 4-hydroxycoumarin derivative and anticoagulant drug included in the World Health Organization's Essential Medicines list[63–65]. Accordingly, we conducted anticoagulant assays, and the biological testing results are presented at the end of this article.

**Scheme 3 |** Scope of bromomethyl arenes[a,b].

| R¹=H, | **27**, 63% | R¹=4-OCHF₂, | **33**, 62% |

R¹=H,        **27**, 63%      R¹=4-OCHF₂,    **33**, 62%
R¹=4-ᵗBu,    **28**, 52%      R¹=4-CH₂OH,    **34**, 51%
R¹=4-Ph (X-ray), **29**, 64%  R¹=4-SO₂Me,    **35**, 62%
R¹=4-F,      **30**, 67%      R¹=4-SO₂NMe₂,  **36**, 65%
R¹=4-CF₃,    **31**, 60%      R¹=3-Cl-4-OMe, **37**, 68%
R¹=4-SCF₃,   **32**, 59%

**38**, 43%

**39**, 55%

**40**, 57%

**41**, 65%

**42**, 67%

**43**, 60%

**44**, 59%
dr = 1:1

**45**, 61%
dr = 1:1

**46**, 64%
dr = 2:1

## Mechanistic studies

To elucidate the possible mechanism of this reaction, we conducted a series of control experiments using the formation of compound 1 as a model reaction (Scheme 6). Initially, we assessed the necessity of the hydroxy group. The experimental results demonstrated that no corresponding product was detected when the hydroxy group was either removed or protected as a benzyl ether (Scheme 6a). Furthermore, when the hydroxy group was protected with TBS and the reaction proceeded, the deprotected product was obtained with an isolated yield of 45% (Scheme 6b). These findings indicate that the hydroxy group is crucial throughout the transformation. Additionally, we investigated the source of protons in the product via isotope labeling experiments. Deuterated DMF and iodobenzene were individually introduced into the reaction system, yet no deuterated product was observed (Schemes 6c, d), suggesting that neither DMF nor iodobenzene acts as a proton source in the transformation. Subsequently, we added varying equivalents of deuterium oxide ($D_2O$) to the reaction system, using dry DMF as the solvent, to examine potential deuterium incorporation into the product (Scheme 6e). The results revealed that in the absence of $D_2O$, the isolated yield of the product significantly decreased to 42%. With the addition of two equivalents of $D_2O$, deuterium incorporation in the product was 5%, with an isolated yield of 74%. Upon adding five equivalents of $D_2O$, deuterium incorporation increased to 20%, and the isolated yield was 53%. Finally, with 10 equivalents of $D_2O$, deuterium incorporation markedly increased to 72%, with an isolated yield of 41%. These results suggest that water in the reaction system may serve as the primary source of additional protons in the product.

Based on the aforementioned control experiments and relevant literature reports[50,66], we propose a possible reaction pathway, using the formation of compound 1 as an example (Scheme 7). Initially, the Ni(II) catalyst is reduced by Mn and coordinates with the substrate to form complex A. This complex then undergoes oxidative addition with iodobenzene, resulting in complex B. Subsequently, complex B is further reduced by Mn, followed by migratory insertion into the olefin to generate the six-membered metallacyclic intermediate D. Finally, intermediate D undergoes protonation to yield product 1 and release the Ni(I) catalyst, which can be reduced by Mn to initiate the next catalytic cycle.

## Biological evaluation of 3-functionalized 4-hydroxycoumarin derivatives

As mentioned above, compounds 58–67 share the same scaffold to warfarin. Consequently, we conducted anticoagulant experiments to assess their biological activity. For the in vivo anticoagulation assay, male ICR (CD-1) mice (6–8 weeks old) were randomly divided into groups of three animals each ($n = 3$). Test compounds (58–67) were administered intravenously at a dose of 20 mg/kg once daily for two consecutive days, while warfarin sodium was used as a reference at 5–20 mg/kg. Bleeding time and bleeding volume were measured 24 h after the last dose. All procedures were conducted in accordance with institutional animal welfare guidelines and approved by the relevant ethics committee (Ethics code: 24-019). Data are presented as mean ± standard deviation (SD), and statistical significance was determined using two-sided one-way ANOVA (GraphPad Prism), with $p < 0.05$ considered significant.

The dosing regimen and blood sampling timeline for the bleeding model are illustrated in Fig. 1(a). The anticoagulant efficacy of the test

**Scheme 4 | Scope of alkenes[a,b].**

R[1]=4-Me-5-Me, **47**, 69%
R[1]=4-OMe, **48**, 57%
R[1]=4-Cl, **49**, 73%
R[1]=5-F, **50**, 65%

**51**, 69%

**52**, 75%

**53**, 67%

**54**, 82%

**55**, 69%

**56** (X-ray), 61%

**57**, 71%

compounds (58–67) was assessed after two days of continuous intravenous (I.V.) administration in mice, based on tail bleeding time and blood loss, as summarized in Figs. 1b, c. One-way ANOVA revealed significant differences among the groups for both bleeding time ($F_{(11, 30)} = 11.17$, $p < 0.0001$, $R^2 = 0.8038$) and blood loss ($F_{(11, 30)} = 10.51$, $p < 0.0001$, $R^2 = 0.7940$). The results indicated that all test compounds, except for compounds 60, 61, and 67, exhibited prolonged bleeding times compared to the vehicle control group (9 minutes), with values ranging from 16 to 45 min. Notably, compounds 59 and 63 demonstrated the most pronounced anticoagulant effects, with bleeding times of 42 minutes (mean difference vs. vehicle: -32.78 min; 95% CI: -46.48 to -19.07; adjusted $p < 0.001$) and 45 minutes (mean difference vs. vehicle: -35.78 min; 95% CI: -49.48 to -22.07; adjusted $p < 0.001$), respectively. Both compounds exhibited greater anticoagulant activity than warfarin, which produced a bleeding time of 29 min. In addition to bleeding time, compounds 59 and 63 also led to a significant increase in blood loss, with values of 27.63 g/kg (mean difference vs. vehicle: -22.95; 95% CI: -33.83 to -12.06; adjusted $p < 0.001$) and 25.97 g/kg (mean difference vs. vehicle: -21.29; 95% CI: -32.17 to -10.40; adjusted $p < 0.001$), respectively, compared to the vehicle control group (4.68 g/kg). These results suggest that compounds 59 and 63 were promising anticoagulant candidates with efficacy superior to that of warfarin.

Additionally, we evaluated the plasma protein binding rates of compounds 58–67 in rat models and observed binding rates exceeding 99.5% (Supplementary Table S5), consistent with that of warfarin[67,68]. This high degree of plasma protein binding reflects the structural and pharmacokinetic similarities between the test compounds and warfarin, thereby validating the use of the in vivo anticoagulant model for comparing their anticoagulant effects.

## Discussion and conclusions
In summary, we have successfully developed a Ni-catalyzed $\beta$-arylation/benzylation of 2'-hydroxychalcone, achieving high chemoselectivity and broad substrate tolerance. The success of this reaction is attributed to the precise interaction between the substrate's intrinsic hydroxyl group and the

Ni catalyst, enabling access to chalcone derivative structures, which may serve as an important manufacturing method in the synthesis of biomedical products and functional materials containing chalcone. Additionally, we demonstrated an application of this practical synthetic approach to facilitate the efficient synthesis of a diverse range of 3-functionalized 4-hydroxycoumarin derivatives with potential anticoagulant activity. Notably, two of the newly synthesized compounds, 58 and 63, exhibited excellent in vivo anticoagulant properties, surpassing those of the clinically used drug warfarin. These findings underscore the utility of this synthetic method in medicinal chemistry and open new avenues for the development of next-generation anticoagulant therapeutics.

Regarding practicality and scalability, the reaction proceeds efficiently on a 4.5 g scale, demonstrating good applicability for larger-scale synthesis. Although the reaction temperature of 110 °C is relatively high compared to ambient conditions, such temperatures are commonly employed in the chemical industry, particularly when using high-boiling solvents like DMF (boiling point 153 °C). Industrial reactors are routinely equipped to safely and efficiently operate at these temperatures, and the use of NiI$_2$, Mn, and DMF is well established, with standard protocols for handling and waste management. As such, the reaction conditions do not pose significant challenges for scale-up or industrial application.

Sustainability and environmental impact remain important considerations in modern synthetic methodology. Ongoing research should be focused on exploring alternative catalytic systems and greener solvents or reductants to further improve the environmental profile and energy efficiency of this transformation. These efforts are expected to broaden the applicability of the method and address environmental considerations in future developments.

Furthermore, in comparison to recent ligand-enabled nickel or palladium-catalyzed hydroarylation methods—which often require external ligands to address challenges, such as $\beta$-H elimination and substrate pre-functionalization—our approach offers distinct advantages. In our system, the substrate's hydroxyl group serves as an intrinsic coordinating site, allowing the substrate to function as both the reaction

**Scheme 5 | Synthetic applications. a** Gram-scale synthesis of 2'-hydroxy dihydrochalcone derivatives. **b** Cyclization of 2'-hydroxy dihydrochalcone derivatives to afford 3-functionalized 4-hydroxycoumarin derivatives as warfarin analogs. **c** One-step synthesis of chroman derivatives from 2'-hydroxy dihydrochalcone derivatives. Reaction conditions: **a** alkene (4.50 g), iodobenzene (3.0 equiv), Mn (3.0 equiv),

NiI$_2$ (10 mol%), P(3-F-Ph)$_3$ (22 mol%), DMF (120 mL), 110 °C, 16 h. **b** phenol (0.2 mmol), diethyl carbonate (4.0 eq), NaH (8.0 eq), o-xylene (3.0 ml), 155 °C, 2–4 h. **c** phenol (0.2 mmol), CH$_2$Br$_2$ (3.0 eq), Cs$_2$CO$_3$ (3.0 eq), o-xylene (3.0 ml), 140 °C, 6 h.

**Scheme 6 | Mechanistic study via control experiments. a** No product formation observed in the absence of the free 2'-hydroxy group. **b** Hydrolyzable protection of the 2'-hydroxy group (TBS) allows product formation. **c** Control experiments with deuterated solvent or **d** reagent indicate that neither the solvent nor the substrate serves as the proton source. **e** Results suggest that water is the primary proton source incorporated into the product.

## Methods

### General procedure for the synthesis of compounds 1–57

A 25 mL Schlenk-type tube equipped with a magnetic stir bar was charged with alkene (0.5 mmol), manganese powder (Mn, 1.25 mmol, 2.5 equiv), nickel(II) iodide (NiI₂, 10 mol%), and tris(3-fluorophenyl)phosphine [P(3-F-Ph)₃, 22 mol%]. The tube was evacuated and backfilled with nitrogen ten times. Subsequently, iodobenzene (1.5 mmol, 3.0 equiv) dissolved in DMF (4 mL) was added via syringe. The reaction mixture was stirred at 110 °C for 16 h. After cooling to room temperature, the mixture was quenched with water and extracted with ethyl acetate. The combined organic layers were washed with brine, dried over anhydrous Na₂SO₄, and concentrated under reduced pressure. The crude product was purified by column chromatography on silica gel (eluent: *n*-hexane/ethyl acetate = 100:1, v/v) to afford the desired product.

### Plasma stability assay

The stability of test compounds (58–67) in rat plasma was evaluated by incubating each compound at a concentration of 2 μg/mL with 100 μL of freshly prepared rat plasma. Samples were incubated at 37 °C with shaking at 80 rpm for 4 h. The reaction was terminated by adding methanol (MeOH) at a 4:1 volume ratio (MeOH:plasma). The remaining concentrations of the test compounds were quantified using LC-MS/MS. Plasma stability was expressed as the percentage of compound remaining at 4 h relative to the initial concentration measured at 0 min.

### Rapid equilibrium dialysis (RED) assay for plasma protein binding

Plasma protein binding (PPB) of test compounds (58–67) was determined using the Rapid Equilibrium Dialysis (RED) method with RED inserts (Catalog No. 90006, Thermo Fisher Scientific), following the manufacturer's protocol. Each compound was evaluated at a concentration of 2 μg/mL at a single time point (4 h) in triplicate. Caffeine (5 μM) and verapamil (5 μM) were included as benchmark controls.

For each assay, 100 μL of rat plasma spiked with the test compound was added to the sample chamber of the RED device, while 350 μL of phosphate-buffered saline (PBS, pH 7.4) was added to the buffer chamber. The RED device was incubated at 37 °C with shaking at 80 rpm for 4 h to allow equilibrium to be reached. After incubation, 5 μL of plasma was collected from the sample chamber and diluted with 195 μL of blank PBS. Simultaneously, 19 μL of buffer was collected from the buffer chamber and diluted with 1 μL of blank plasma. The diluted samples were vortexed at room temperature for 5 min. Subsequently, 20 μL of each diluted sample was mixed with 80 μL of methanol, followed by centrifugation at 4500 rpm for 25 min. The resulting supernatants were collected and analyzed by LC-MS/MS.

The fraction bound ($f_{bound}$) of each compound to plasma proteins was calculated as follows:

$$f_{bound} = 1 - f_{unbound}$$

**Scheme 7 |** Plausible mechanism.

where the unbound fraction ($f_{unbound}$) was determined by:

$$f_{unbound} = C_{buffer}/(C_{buffer} + C_{plasma})$$

Here, $C_{buffer}$ represents the concentration of the test compound in the buffer chamber, and $C_{plasma}$ represents the concentration in the plasma chamber. The percentage of PPB was calculated using the formula:

$$PPB(\%) = (1 - f_{unbound}) \times 100\%$$

**In Vivo anticoagulation assay**

Male ICR (CD-1) mice (6–8 weeks old) were randomly divided into groups ($n = 3$ mice per group). Test compounds (58–67) were administered via intravenous injection (I.V.) once daily for two consecutive days (Day 0 and Day 1) at a dose of 20 mg/kg. Warfarin sodium was used as a reference compound at a dose of 5–20 mg/kg (no difference in efficacy observed across this range).

On Day 2 (24 h after the last dose), mice were anesthetized by intraperitoneal injection (I.P.) of a fentanyl/midazolam/medetomidine mixture (0.05, 5, and 0.5 mg/kg, respectively). After confirming adequate anesthesia, an incision was made 15 mm from the tail tip. The animals were placed on a heat pad, and the snipped tails were hung vertically. Blood was collected in a 1.5 mL Eppendorf tube for up to 60 min, or until bleeding stopped or 40% of the total blood volume was reached. Bleeding time was defined as the time elapsed until bleeding ceased, and bleeding volume was determined by weighing the collected blood after bleeding had stopped. At the end of the experiment, all mice were euthanized by $CO_2$ inhalation followed by cervical dislocation.

**Ethics Statement**

All the procedures related to animal welfare in this study are in compliance with the animal welfare policies and the guidelines of Drug Safety Testing Center (DSC), Hong Kong. The study has been reviewed and approved by the DSC Institutional Animal Care and Use Committee (IACUC) (Ethics code: 24-019).

**Reporting summary**

Further information on research design is available in the Nature Portfolio Reporting Summary linked to this article.

**Data availability**

The authors declare that the data supporting the findings of this study are available within the paper and its Supplementary Information. Including experimental procedures for chemistry (Supplementary Method 1), product characterization (Supplementary Note 4), crystallographic data (CIF, Supplementary Note 5, Figures S1-S5), materials and methods for biological study (Supplementary Notes 6-8, Supplementary Methods 2-8), [1]H, [13]C and [19]F NMR spectra (Supplementary Notes 3 and 9). The X-ray crystallographic data for compounds 1, 8, 29, 56 and 58 reported in this study has been deposited to the Cambridge Crystallographic Data Center (CCDC: 2267471, 2359824, 2359825, 2369868 and 2401627, respectively). This data can be obtained free of charge via CCDC at: https://www.ccdc.cam.ac.uk/data_request/cif.

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

## Acknowledgements

We gratefully acknowledge the financial support from the Research Grants Council of the Hong Kong Special Administrative Region, China (PolyU 15100022 to C.M.), Hong Kong Polytechnic University (State Key Laboratory of Chemical Biology and Drug Discovery, and Marshall Research Center for Medical Microbial Biotechnology to C.M.), and the Chinese University of Hong Kong (Passion for Perfection Scheme PFP202210-008 to X.Y.). We thank the University Research Facility in Life Sciences (ULS) of the Hong Kong Polytechnic University for the technical assistance.

## Author contributions

B.-C.T. and C.M. conceived the idea. X.Y. and C.M. supervised the project and acquired the funding. B.-C.T., X.S., and J.L. conducted the laboratory work. B.-C.T., X.S., X.Y., and C.M. analyzed the data. B.-C.T., X.S., X.Y., and C.M. wrote the manuscript.

## Competing interests

The authors declare no competing interests.
