## [Transparent Peer review file · Communications Chemistry]

Nickel-catalyzed β -arylation and benzylation of 2'-hydroxychalcones to access warfarin analogues

Corresponding Author: Professor Cong Ma

Version 0:

Reviewer comments:

Reviewer #1

(Remarks to the Author)

Recommendation: Receive after revisions noted.

The article described a Ni-catalyzed β -arylation/benzylation of 2'-hydroxy-chalcone, leading to the synthesis of novel chalcone derivatives. To demonstrate the utility of this synthetic approach, author converted the chalcone derivatives into a series of 3-functionalized 4-hydroxycoumarin derivatives through a one-step transformation. These derivatives exhibited excellent anticoagulant effects in vivo, with compounds 59 and 63 showing anticoagulant activity superior to that of warfarin. This method holds significant promise for the development of new therapeutic agents and functional materials featuring the chalcone structure. Therefore I would suggest the acceptance of this manuscript after some revisions.

1. The Greek letters in the article should be italic.
2. In the substrate expansion, whether the benzyl bromide substrate can extend the carbon chain to become an alkyl bromide?
3. Mn as a reducing agent, how about the effect of other reducing agents?
4. Has the author tried any directing groups except phenolic group?
5. In the mechanism circle, intermediate D undergoes protonation reaction to obtain product 1. What provides proton hydrogen?
6. There are several issues that can be corrected prior to publication.
 - Supporting Information: In the optimization of reaction conditions, NiI₂ and Ni(OTf)₂ have almost the same effect, and whether there is a reverse screening to determine the final effect?
 - Supporting Information: In the General procedure, P(3-F-Ph)₃ (22 mmol%)  P(3-F-Ph)₃ (22 mol%)
 - Supporting Information: For the following data, the J-coupling should be changed to the italic format: 2, 3, 9, 20, 21, 22, 23, 24, 27, 28.

Reviewer #2

(Remarks to the Author)

The authors developed a Ni-catalysed reductive coupling of haloarenes and 2'-hydroxy-chalcones, resulting in β -arylation/benzylation using the 2-hydroxy group as a directing group and iodobenzene or benzyl bromide derivatives as the arylation and benzylation source, respectively. The use of 2-hydroxy group as a directing group resulted in excellent selectivity and after optimization, the use of P(3-F-Ph)₃ as a ligand was shown to improve the yield significantly. The authors also showed bioactivity of the products they synthesised being biologically relevant, one as a better anti-coagulant and similar plasma protein binding rates when compared to Warfarin, an anticoagulant drug listed in the World Health Organization's Essential Medicines. Mechanistic studies were performed to elucidate the mechanism, and the authors proposed a reductive coupling mechanism, with the proton most likely arising from trace water. I recommend publication in Communications Chemistry after the following issues are addressed:

1. The manuscript's flow should be revised. Currently, it shifts from chemistry to biology and then back to chemistry. The biological data should be moved to the end to ensure better narrative consistency. This change should also be reflected in the Supplementary Information (SI).
2. The scope of the reaction with α - and β -substituted chalcones was not explored. Substitution at the β -position is

particularly interesting, as it can generate a potential stereogenic center. This part should be expanded in the paper.

3. In Table 1, the term "isolated yield based on 2'-hydroxychalcone" is unclear. Isolated yield should be reported based on the final product (e.g. Product 1), not the starting material. The current wording implies yield was calculated based on recovered starting material, which would be inaccurate. The same clarification is needed in Scheme 2, where yields are reported based on the alkene.

4. In the mechanistic studies, did the authors consider alternative directing groups beyond OH? Compounds 16, 17, 40, and 42 suggest that other functional groups (e.g., N-H, S-containing groups) may be tolerated. Broadening the range of viable directing groups would enhance the utility of the methodology.

5. Given the importance of the protonation step and the observed increase in yield with increasing equivalents of D₂O, did the authors try direct addition of H₂O to improve yields? While most substrates give good yields, some (e.g. 38, 39) are less efficient. The authors did attempt using 1.5 equiv of H₂O during the optimization stage, but that was without the ligand, which led to poor yields. Considering the increase from 42% to 74% yield with the addition of 2 equiv of D₂O, this is a promising direction to explore.

6. The capitalization of chemical elements, such as nickel, is inconsistent throughout the manuscript. In some cases, elemental names are capitalized; in others, their periodic symbols (e.g. Ni) are used. Either approach is fine, but consistency is necessary throughout the manuscript.

7. Words such as "entry" and "entries" are inconsistently capitalized. Please ensure consistent formatting of such terms throughout the text.

Reviewer #3

(Remarks to the Author)

This manuscript describes a nickel-catalyzed β -arylation and benzylation of 2'-hydroxychalcone, enabled by the substrate's intrinsic hydroxyl group. The authors report excellent chemoselectivity, a broad substrate scope, and compelling biological validation of their products as anticoagulant agents, with two compounds outperforming warfarin *in vivo*. This work presents a synthetically valuable and biologically relevant advance and is generally well-suited for publication in *Communications Chemistry*, pending minor revisions.

Major Strengths

Conceptual originality: The use of a native hydroxy group for directing the transformation without the need for external auxiliaries is a significant contribution to the field of substrate-directed catalysis.

Broad synthetic scope: The reaction is shown to be general across a variety of aryl and benzyl coupling partners, including challenging and functionalized substrates.

Biological impact: The discovery of novel anticoagulant compounds with greater efficacy than warfarin gives the synthetic chemistry clear medicinal relevance.

Mechanistic insight: The authors conduct thoughtful mechanistic experiments to support the substrate-directed nature of the transformation and the role of water as the proton source.

Data quality: Characterization data, crystal structures, and biological testing are comprehensive and well-documented.

Suggestions

Mechanistic Clarity:

The mechanistic scheme (Scheme 7) could be improved by clearly showing oxidation states and transition states/intermediates (e.g., complex A to D).

The role of manganese in the reduction cycle should be better explained.

Comparative Discussion:

The discussion would benefit from a more detailed comparison with recent ligand-enabled nickel or palladium-catalyzed hydroarylations and how this method circumvents typical limitations (e.g., β -H elimination, pre-functionalization).

Reaction Conditions

Contextualization:

Briefly address the practicality and environmental considerations of the conditions used (e.g., use of NiI₂, Mn, DMF, and elevated temperatures), especially in light of potential scalability.

Biological Methodology Transparency:

The main text should briefly summarize key experimental details (e.g., number of animals, statistical analysis methods), not just defer entirely to the Supplementary Information.

Graphical Enhancements:

A graphical abstract or a simplified reaction overview (suitable for TOC or social media promotion) would improve visibility and accessibility.

Minor Points

Consistency in formatting (Greek characters, temperatures, spacing) should be double-checked.

Adding more recent citations (2022–2024) on nickel-catalyzed hydroarylation and C–H activation would further strengthen the introduction.

Minor Revision

This manuscript makes a substantial contribution to both synthetic methodology and bioactive compound discovery. I recommend publication after the authors address the above points to improve clarity, depth, and accessibility.

Version 1:

Reviewer comments:

Reviewer #2

(Remarks to the Author)

The authors have fully addressed my comments. I recommend acceptance of the paper.

Reviewer #3

(Remarks to the Author)

The revised version of the manuscript can be published.

Responses addressing reviewers' comments

Reviewer #1

1. *The Greek letters in the article should be italic.*

Thank you for the comment. The modifications have been made accordingly.

2. *In the substrate expansion, whether the benzyl bromide substrate can extend the carbon chain to become an alkyl bromide?*

Thank you for your valuable suggestion. We have tested a series of alkyl bromides under the optimized reaction conditions. Unfortunately, these experiments resulted in the formation of highly complex mixtures, rather than the desired products. This outcome suggests that the current protocol may not be suitable for alkyl bromides, possibly due to competing side reactions or lower selectivity. This information has been supplemented into the revised manuscript as follows:

“We have also tested a series of alkyl bromides. Unfortunately, these experiments resulted in the formation of highly complex mixtures, rather than the desired products. This outcome suggests that the current protocol may not be suitable for alkyl bromides, possibly due to competing side reactions or lower selectivity.”

3. *Mn as a reducing agent, how about the effect of other reducing agents?*

Thank you for your suggestion. We also explored other types of reducing reagents, including Zn, HSiEt₃, PhSiH₃, and HCOOK as described in SI. However, none of these alternatives provided a higher yield than Mn under our reaction conditions. This information has been supplemented into the revised manuscript as follows:

“Other types of reducing reagents were also explored, including Zn, HSiEt₃, PhSiH₃, and HCOOK. However, none of these alternatives provided a higher yield than Mn (see SI for more details).”

4. *Has the author tried any directing groups except phenolic group?*

In our initial investigations, we also evaluated other directing groups, including pyridine, primary amine, and secondary amine. However, none of these alternatives were effective in promoting the desired reaction under our conditions.

5. *In the mechanism circle, intermediate D undergoes protonation reaction to obtain product 1. What provides proton hydrogen?*

Based on our control experiments using D₂O, it's reasonable to conclude that the additional proton primarily originates from external water.

6. There are several issues that can be corrected prior to publication.
•Supporting Information: In the optimization of reaction conditions, NiI_2 and $\text{Ni}(\text{OTf})_2$ have almost the same effect, and whether there is a reverse screening to determine the final effect?

Thank you for your suggestion. We conducted the reverse screening as recommended and found that NiI_2 still performed slightly better than $\text{Ni}(\text{OTf})_2$ under our reaction conditions.

•Supporting Information: In the General procedure, $\text{P}(3\text{-F-Ph})_3$ (22 mmol%)  $\text{P}(3\text{-F-Ph})_3$ (22 mol%)

Thank you for pointing out this typo. "22 mmol%" has been corrected to "22 mol%" in the revised SI.

•Supporting Information: For the following data, the J-coupling should be changed to the italic format: 2, 3, 9, 20, 21, 22, 23, 24, 27, 28.

Thank you for pointing out the formatting issue. The format of these J-couplings has been revised to italics in SI.

Reviewer #2

1. The manuscript's flow should be revised. Currently, it shifts from chemistry to biology and then back to chemistry. The biological data should be moved to the end to ensure better narrative consistency. This change should also be reflected in the Supplementary Information (SI).

Thank you for the suggestion. We have moved the biological section to the end and made the corresponding changes in both the main text and the SI.

2. The scope of the reaction with α - and β -substituted chalcones was not explored. Substitution at the β -position is particularly interesting, as it can generate a potential stereogenic center. This part should be expanded in the paper.

Thank you very much for your insightful suggestion. We agree that α - and β -substituted chalcones are highly attractive substrates for this addition reaction and could provide valuable insights. However, the synthesis of these substituted chalcones often involves multi-step procedures and would require considerable time and effort, which is beyond the current scope of our study. Additionally, the presence of α - or β -substituents may introduce additional steric or electronic effects, potentially complicating the reaction outcome and mechanistic interpretation. Given these considerations, and in light of our aim to establish the generality of this new

reaction using readily available substrates, we believe that our manuscript presents a complete and coherent story as it stands. We consider the exploration of α - and β -substituted chalcones to be an excellent direction for future research.

3. In Table 1, the term “isolated yield based on 2'-hydroxychalcone” is unclear. Isolated yield should be reported based on the final product (e.g. Product 1), not the starting material. The current wording implies yield was calculated based on recovered starting material, which would be inaccurate. The same clarification is needed in Scheme 2, where yields are reported based on the alkene.

We apologize for any confusion caused by the previous wording. The table note has been revised to “Isolated yield.”

4. In the mechanistic studies, did the authors consider alternative directing groups beyond OH? Compounds 16, 17, 40, and 42 suggest that other functional groups (e.g., N-H, S-containing groups) may be tolerated. Broadening the range of viable directing groups would enhance the utility of the methodology.

In our initial investigations, we also evaluated other directing groups, including pyridine, primary amine, and secondary amine. However, none of these alternatives were effective in promoting the desired reaction under our conditions.

5. Given the importance of the protonation step and the observed increase in yield with increasing equivalents of D_2O , did the authors try direct addition of H_2O to improve yields? While most substrates give good yields, some (e.g. 38, 39) are less efficient. The authors did attempt using 1.5 equiv of H_2O during the optimization stage, but that was without the ligand, which led to poor yields. Considering the increase from 42% to 74% yield with the addition of 2 equiv of D_2O , this is a promising direction to explore.

Thank you for your suggestion. Indeed, water plays an important role in this transformation. We attempted to add additional water under the optimal reaction conditions; however, this did not further improve the yield. Our results indicate that a small amount of water is optimal, as neither using super-dry solvent nor adding extra water to regular solvent increased the yield. This observation also explains why adding 2 equivalents of water to super-dry DMF led to an improved yield.

6. The capitalization of chemical elements, such as nickel, is inconsistent throughout the manuscript. In some cases, elemental names are capitalized; in others, their periodic symbols (e.g. Ni) are used. Either approach is fine, but consistency is necessary throughout the manuscript.

Thank you for the comment. We have made the corresponding changes and have used element symbols consistently throughout the revised manuscript.

7. Words such as “entry” and “entries” are inconsistently capitalized. Please ensure consistent formatting of such terms throughout the text.

Thank you for the comment. We have made the corresponding changes and have used the term of “Entry” and “Entries” consistently throughout the revised manuscript.

Reviewer #3

1. The mechanistic scheme (Scheme 7) could be improved by clearly showing oxidation states and transition states/intermediates (e.g., complex A to D). The role of manganese in the reduction cycle should be better explained.

Thank you for the suggestion. We have added the oxidation states to the scheme and included a discussion on the role of Mn in the revised manuscript as follows:

“Based on the aforementioned control experiments and relevant literature reports,^{47, 65} we propose a possible reaction pathway, using the formation of compound **1** as an example (Scheme 7). Initially, the Ni(II) catalyst is reduced by Mn and coordinates with the substrate to form complex **A**. This complex then undergoes oxidative addition with iodobenzene, resulting in complex **B**. Subsequently, complex **B** is further reduced by Mn, followed by migratory insertion into the olefin to generate the six-membered metallacyclic intermediate **D**. Finally, intermediate **D** undergoes protonation to yield product **1** and release the Ni(I) catalyst, which can be reduced by Mn to initiate the next catalytic cycle.

Scheme 7. Plausible mechanism.

2. The discussion would benefit from a more detailed comparison with recent ligand-enabled nickel or palladium-catalyzed hydroarylations and how this method circumvents typical limitations (e.g., β -H elimination, pre-functionalization).

Thank you for your valuable suggestion. Unlike traditional ligand-enabled nickel or palladium-catalyzed hydroarylation methods, which often require external ligands to facilitate catalysis and circumvent challenges such as β -H elimination and substrate pre-functionalization, our approach leverages the substrate's intrinsic hydroxyl group as a coordinating site. This unique feature allows the substrate to act as both the reaction center and the coordination center, effectively promoting hydroarylation without the need for additional ligands. As a result, our method addresses typical limitations associated with β -H elimination and pre-functionalization, offering a more straightforward and efficient catalytic process. We have expanded the discussion in the revised manuscript as follows to highlight these distinctions and the advantages of our strategy:

“Furthermore, in comparison to recent ligand-enabled nickel or palladium-catalyzed hydroarylation methods—which often require external ligands to address challenges such as β -H elimination and substrate pre-functionalization—our approach offers distinct advantages. In our system, the substrate's hydroxyl group serves as an intrinsic coordinating site, allowing the substrate to function as both the reaction center and the coordination center. This unique feature eliminates the need for additional ligands and effectively circumvents typical limitations associated with β -H elimination and pre-functionalization. As a result, our method provides a more straightforward and efficient catalytic process. Collectively, these advances

demonstrate the broad applicability and potential impact of our methodology in both synthetic and medicinal chemistry.”

3. Briefly address the practicality and environmental considerations of the conditions used (e.g., use of Ni_2 , Mn, DMF, and elevated temperatures), especially in light of potential scalability.

Thank you for your insightful comment regarding the reaction temperature. We acknowledge that 110 °C is relatively high compared to ambient conditions. However, this temperature was found to be necessary to achieve optimal yields and selectivity under our current reaction conditions. It is important to note that in the chemical industry, reaction temperatures in the range of 100–150 °C are quite common, particularly when using high-boiling solvents such as DMF (boiling point 153 °C). Industrial reactors are routinely equipped to safely and efficiently operate at these temperatures, and such conditions do not pose significant challenges for scale-up or practicality.

Furthermore, Ni_2 , Mn, and DMF are all standard catalyst, reagent and solvent in the chemical industry, with well-established protocols for handling and waste management. We have also demonstrated that the reaction can be performed efficiently on a 4.5 g scale, supporting its practicality and scalability.

Nonetheless, we recognize the importance of developing more sustainable and energy-efficient processes. As part of our ongoing research, we are actively investigating alternative catalytic systems and reaction conditions that could enable this transformation at lower temperatures or with greener reagents and solvents. We believe these efforts will further enhance the environmental profile and industrial applicability of our methodology.

This information has been added into the revised manuscript as follows:

“Regarding practicality and scalability, the reaction proceeds efficiently on a 4.5 g scale, demonstrating good applicability for larger-scale synthesis. Although the reaction temperature of 110 °C is relatively high compared to ambient conditions, such temperatures are commonly employed in the chemical industry, particularly when using high-boiling solvents like DMF (boiling point 153 °C). Industrial reactors are routinely equipped to safely and efficiently operate at these temperatures, and the use of Ni_2 , Mn, and DMF is well established, with standard protocols for handling and waste management. As such, the reaction conditions do not pose significant challenges for scale-up or industrial application.

Sustainability and environmental impact remain important considerations in modern synthetic methodology. Ongoing research **should be** focused on exploring alternative catalytic systems and greener solvents or reductants to further improve the environmental profile and energy efficiency of this transformation. These efforts are expected to broaden the applicability of the method and address environmental considerations in future developments.”

4. *Biological Methodology Transparency:*

The main text should briefly summarize key experimental details (e.g., number of animals, statistical analysis methods), not just defer entirely to the Supplementary Information.

Thank you for the valuable comment. We have summarized the experimental details and added into the revised manuscript as follows:

“For the *in vivo* anticoagulation assay, male ICR (CD-1) mice (6–8 weeks old) were randomly divided into groups of three animals each (n = 3). Test compounds (58–67) were administered intravenously at a dose of 20 mg/kg once daily for two consecutive days, while warfarin sodium was used as a reference at 5–20 mg/kg. Bleeding time and bleeding volume were measured 24 hours after the last dose. All procedures were conducted in accordance with institutional animal welfare guidelines and approved by the relevant ethics committee (Ethics code: 24-019). Data are presented as mean ± standard deviation, and statistical significance was determined using one-way ANOVA (GraphPad Prism), with p < 0.05 considered significant.”

5. *A graphical abstract or a simplified reaction overview (suitable for TOC or social media promotion) would improve visibility and accessibility.*

We have supplemented a graphical abstract with the revised manuscript for submission.

6. *Consistency in formatting (Greek characters, temperatures, spacing) should be double-checked.*

Thank you for bringing this to our attention. We have thoroughly reviewed the manuscript for formatting consistency and have implemented the necessary revisions to ensure uniformity.

7. *Adding more recent citations (2022–2024) on nickel-catalyzed hydroarylation and C–H activation would further strengthen the introduction.*

Thank you for your valuable suggestion. We have cited three recent reports on Ni-catalyzed ligand-enabled hydroarylations as follows to provide a more comprehensive comparison:

26. X. Chen, Y. Wu, R. Zhang, F. Wang and C. Chen, Regioselective Nickel-Catalyzed Hydroarylation of gem-Difluoroalkenes for the Synthesis of the ArCF₂- Moiety, *Angew Chem Int Ed Engl*, 2025, **64**, e202424714.

27. P. Mukherjee, Z. Alassad and T. K. Hyster, Synergistic Photoenzymatic Anti-Markovnikov Hydroarylation of Olefins via Heteroaryl Radical Intermediates, *J Am Chem Soc*, 2025, **147**, 14048-14053.
28. Z. C. Wang, L. Gao, S. Y. Liu, P. Wang and S. L. Shi, Facile Access to Quaternary Carbon Centers via Ni-Catalyzed Arylation of Alkenes with Organoborons, *J Am Chem Soc*, 2025, **147**, 3023-3031.

We would like to take this opportunity to thank the editor and all the reviewers for your valuable comments and suggestions, which have greatly helped us to improve the quality of this manuscript.